# Influences of hand action on the processing of symbolic numbers: A special role of pointing?

**Mariagrazia Ranzini** [ID] [1]*, **Carlo Semenza**[2], **Marco Zorzi**[1,3], **Simone Cutini**[4]

**1** Department of General Psychology (DPG), University of Padua, Padua, Italy, **2** Department of Neuroscience (DNS), University of Padua, Padua, Italy, **3** IRCCS San Camillo Hospital, Venice-Lido, Italy, **4** Department of Developmental Psychology and Socialisation (DPSS), University of Padua, Padua, Italy

* mariagrazia.ranzini@unipd.it

**Data Availability Statement:** The anonymised dataset associated to this article are available on the Open Science Framework (OSF) https://osf.io/a8jzp/.

## Abstract

Embodied and grounded cognition theories state that cognitive processing is built upon sensorimotor systems. In the context of numerical cognition, support to this framework comes from the interactions between numerical processing and the hand actions of reaching and grasping documented in skilled adults. Accordingly, mechanisms for the processing of object size and location during reach and grasp actions might scaffold the development of mental representations of numerical magnitude. The present study exploited motor adaptation to test the hypothesis of a functional overlap between neurocognitive mechanisms of hand action and numerical processing. Participants performed repetitive grasping of an object, repetitive pointing, repetitive tapping, or passive viewing. Subsequently, they performed a symbolic number comparison task. Importantly, hand action and number comparison were functionally and temporally dissociated, thereby minimizing context-based effects. Results showed that executing the action of pointing slowed down the responses in number comparison. Moreover, the typical distance effect (faster responses for numbers far from the reference as compared to close ones) was not observed for small numbers after pointing, while it was enhanced by grasping. These findings confirm the functional link between hand action and numerical processing, and suggest new hypotheses on the role of pointing as a meaningful gesture in the development and embodiment of numerical skills.

## Introduction

In everyday life, humans make use of the hands to deal with numbers in a variety of contexts. Indeed, the hands are used to visually represent numbers, to perform arithmetic calculations, to count series of items, as well as to communicate magnitude information [1–4]. The use of the hands in numerical processing appears to be long-standing universally shared practice, since it has been observed nowadays as well as in ancient civilisations [2, 5]. Many studies indicate that the use of fingers for counting and arithmetic is a cornerstone for the understanding and development of number concepts [6, 7].

The link between numbers and hands has often been discussed in the context of embodied and grounded theories of cognition, which suggest that high level cognitive processes develop

**Funding:** This study was carried out within the scope of the research program Dipartimenti di Eccellenza (art.1, commi 314-337 legge 232/2016), which was supported by a grant from MIUR to the Department of General Psychology, University of Padua. Also, this work was funded by the European Union's Horizon 2020 research and innovation program under Marie Skłodowska-Curie Grant 839394 (to M.R.). The funders had no role in study design, data collection and analysis, decision to publish, or preparation of the manuscript.

**Competing interests:** The authors have declared that no competing interests exist.

through sensorimotor experience [8, 9]. Grounded, embodied and situated aspects of cognition are distinct [10]: grounded aspects concern the fact that sensorimotor experiences and the development of cognitive skills or mental representations might be determined by universal laws (e.g., direction of gravity). Embodied aspects refer to body movements during repeated cognitive tasks (e.g., cultural habits), creating long-lasting links between sensorimotor and cognitive processes. Situated aspects concern features of a specific environmental context, which creates contingent links between sensorimotor and cognitive processes (e.g., task requirements; see also: [11]).

In the context of numerical cognition, culturally-specific practices associated to the use of hands for counting and the promotion of these practices at school can be seen as *embodied* processes. Nonetheless, growing evidence suggests that mental processes related to number and hand actions are not merely determined by cultural habits, because the latter might be built upon cognitive and neural constraints [1, 12]. First, several studies have shown associations between the development of fine motor skills and number knowledge (for a review, see [13]). For instance, finger gnosis (i.e., the ability to mentally represent and discriminate each finger) is predictive of number-related abilities in preschoolers [14], in primary school children [15, 16], and in adults [17]. Similarly, other studies have found that a variety of abilities related to fine hand movements (e.g., drawing, precision grasping, tapping, etc.) predicts performance in number-related tasks in preschoolers [18, 19]. Second, neuroimaging and neuropsychological studies consistently suggest the existence of common neural networks for hand action and numerical processing [20–26; see 27 for a recent metanalysis of neuroimaging studies].

Taken together, these observations support the idea that numerical processing is both embodied on cultural habits and constrained by neurocognitive mechanisms. In line with this idea, an increasing number of behavioural studies have described interactions between symbolic number processing and the hand action of reaching and grasping objects, suggesting overlapping mechanisms between the mental representation of numbers and sensorimotor processes [28, 32–36, 38–41]. Reaching often precedes grasping in many actions of everyday life. Nonetheless, reach to grasp actions are characterised by two distinguishable components: arm transport and hand preshaping [29]. Arm transport consists in moving the hand and arm to reach an object, thereby involving visuospatial mechanisms for processing object location and for online monitoring of the movement trajectory. Hand preshaping for grasping concerns the processes required to plan and select the correct hand configuration for grasping, and it involves object size judgments. Neuroimaging studies have reported anatomical dissociation between the two components within the frontoparietal grasp network, the neural locus of arm transport being located in more posterior parietal areas, while the neural locus of grasp preshaping being located in the anterior part of the intraparietal sulcus [30, 31].

Studies investigating the interplay between number and action have observed interactions both between symbolic number processing and hand reaching, and between symbolic number processing and hand grasping. Concerning hand reaching, previous studies have shown that processing number magnitude or number distance can influence reaching trajectories. For instance, [32] found that leftward or rightward pointing was faster during the processing of small or large digits, respectively. Also, deviation from a standard trajectory occurs as a function of numerical distance [33], and targets consisting of larger digits biases finger trajectories more rightward as compared to smaller digits [34]. Similarly, when required to reach, grasp, and freely displace an object while performing at the same time a comparison of Arabic digits magnitude, participants systematically misplaced the object leftward while processing smaller digits as compared to larger ones [35]. Finally, reach velocity while connecting with the index finger numerical stimuli on a touch screen was higher with more distant numbers than closer

ones (e.g., 1–2 vs. 1–8: [36]). Taken together, these interactions between number and reaching are in line with and support the well-established hypothesis of the *mental number line*, postulating that the mental representation of numbers is spatial in nature, and oriented from left to right, at least in Western societies as a possible consequence of reading and writing habits (e.g., [37]). In this view, smaller numbers facilitate the planning and execution of reaching toward the left side of the space, whereas larger numbers enhance reaching rightwards (see [38, 39], for reviews). Similar results were also observed in tasks requiring the use of different effectors; for instance, the trajectory of ocular movements during numerical processing is coherent with the mental number line [40], suggesting that a common mechanism might exist between numerical processing and action planning and execution which locate reaching direction of actions.

Concerning hand grasping, several studies have observed that number magnitude influences performance in tasks requiring grasping actions. For instance, Andres et al. [41] observed that, in a parity judgment task, participants were faster in initiating a closure or opening grip with smaller or larger digits, respectively. Similarly, follow-up studies found that participants are faster in initiating a precision or power grasp with smaller or larger digits respectively [42, 43]. Lastly, Andres et al. [44] observed that maximal finger aperture during grasping—i.e., the Euclidean distance between index and thumb—was larger when a large digit (e.g., 8) was written on the object to be grasped compared to when the written digit was small (e.g., 2). These findings are in line with the ATOM theory (ATOM: A Theory of Magnitude) first proposed by Walsh [45] and later updated in [46, 47] (see also: [48]). ATOM postulates the existence of a common neurocognitive network where numerical, physical, and temporal magnitudes are commonly represented to support sensorimotor transformations. In this sense, interactions between number and grip size strongly support the ATOM theory as they can be seen as resulting from a common representation of numerical and physical dimensions during action.

Importantly, interactions between number and hand actions are bidirectional: not only numbers bias hand action, but also observing hand actions influences the processing of numbers. Specifically, several studies have shown that observing fingers depicting grip closing impacts the processing of number magnitude [49, 50], and biases random number generation [51] (see also [28]). Action-number interactions in these studies were specific for conditions where grip closing, but not grip opening, was depicted by a human hand, and not by a fake hand, demonstrating the sensorimotor nature of these effects [49] (see also [52] for comparison between biological and non-biological movements). In a similar vein, Ranzini et al. [53] observed that presenting graspable objects, as compared to ungraspable ones, enhanced the sensitivity to numerical magnitude during a subsequent number task (Experiment 1), but it reduced number magnitude sensitivity when participants were additionally required to hold an object in their hands (Experiment 2). Taken together, these bidirectional links suggest a functional relation between motor action and numerical processing, in agreement with embodied and grounded cognition accounts (see also: [54–56]).

In this study we sought to investigate the link between numerical processing and hand action using an experimental setting where hand action and number are processed in distinct tasks. To do this, we designed a paradigm based on motor adaptation. Motor adaptation induces neural adaptation, i.e., the progressive reduction in the neural activity of relevant brain areas following stimulus repetition, a phenomenon which has been consistently useful in neuroscience studies [57, 58]. In neuroimaging studies, neural adaptation paradigms have been mainly used to investigate the functioning of sensory systems (for a review see: [59]), and also to investigate the functioning of brain areas associated to cognitive or motor processing, such as number (e.g., [60]) or processes related to hand grasping (e.g., [61]). In behavioural

studies, some previous studies already took advantage of motor adaptation procedures to investigate the link between cognition and motor action: specifically, repeated hand motor actions have been found to affect the processing of words referring to concrete or abstract concepts [62], and the processing of time, space and numerosity [63]; also effects of hand body posture in a squeezing task has been found to influence the perception of grasp possibilities, as well as the processing of numerical magnitude [64]. Here, for the first time, we used motor adaptation to investigate the link between mechanisms involved in reach and grasp hand movements and symbolic number processing.

In this study, participants performed repetitive grasping of an object (arm transport + grasp preshaping), repetitive pointing to the object (arm transport w/o grasp preshaping), repetitive hand tapping (motor control condition), or passively viewed at the object (motionless control condition). Subsequently, they performed the numerical task, consisting in comparing a target digit with 5 (the reference number). We were interested in the specific effects of adaptation to arm transport and grasp preshaping on number. More precisely, we argue that number and size might be linked by a common system devoted to action for estimating both physical size (e.g., object size) and number magnitude, as predicted by ATOM, whereas number and space might be linked by a common system devoted to action for estimating both physical and numerical spatial locations and distances, as postulated in the mental number line hypothesis. In this sense ATOM and the mental number line hypothesis might highlight different aspects of the way numbers are mentally represented. The results of a recent meta-analysis of neuro-imaging studies also support this view, indicating partially distinct parietal brain areas common to symbolic number processing and grasping vs. reaching actions [27]. Interestingly, a previous study [65] came to a similar conclusion. In their study, participants were required to perform parity judgments while maintaining a numerical sequence in working memory. They found a compatibility between type of grip and numerical magnitude, and a compatibility between order of the item in working memory and side of response, indicating that distinct aspects of the mental representation of numbers should be functionally related to distinct motor actions.

Importantly, as compared to the above-mentioned related studies, our paradigm had several novel aspects. First, hand action and number comparison were functionally and temporally dissociated. Many experimental paradigms used to investigate bidirectionality intermixed numbers and hand actions, and required the participant to consider both stimuli to solve the task (e.g., to indicate the odd number within a pair when a hand closing is seen; [49]). The observed effects often consisted in associations between numerical magnitude—or distance—and movement features, such as facilitation in processing small/large numbers with precision grip/power grasp [42], or faster movements when connecting numerically distant numbers as compared to closer ones [36]. Intermixed hand action and number-related processes might favour effects due to situated aspects, such as the use of implicit associative strategies: for instance, one might create ad-hoc associations between grip closing with small numbers and grip opening with large numbers, without necessarily sharing the cognitive processing of number and motor action (see also [64] for a discussion on this point). To avoid this concern, in our paradigm, participants performed hand action prior to number comparison, and the two tasks were unrelated and without any temporal overlap. Also, we avoid the use of movements which could be easily dichotomously classified (such as power vs. precision grip, or left vs. right movements). In this sense, the strength of our paradigm lies in the absence of any potential characteristics favouring the use of associative strategies, allowing to discuss the results in terms of common cognitive mechanisms between number and hand action. Furthermore, in the present study participants executed hand actions, instead of observing them. Previous

studies have often used visual stimuli to trigger sensorimotor processes, in light of the view that observing actions recruits the same neural mechanisms involved in action execution [66]. In the context of studies on numerical cognition, interactions between observed hand action and numerical processing were indeed observed. However, we argue that the execution of hand actions—instead of passive hand viewing—provides a more ecological setting and a stronger test for the idea of a functional link between hand action and numerical processing. Finally, the use of adaptation paradigms to probe cognitive processes and the underlying neuronal populations (e.g., [57, 59]) assures relatively durable changes in the neural mechanisms involved, with the consequence that biases on numerical processing should be safely attributed to the experimental manipulation.

We tested two specific hypotheses:

- hypothesis 1 (HP1): functional overlap between the mechanisms of hand reaching /grasping actions and those involved in numerical processing predicts that the latter should be affected by motor adaptation: specifically, repetitive reaching (i.e., arm transport common to grasp and point actions) and grasping (specific of the grasp action) would affect the speed in judging numerical magnitude (i.e., number comparison) as compared to tapping and passive viewing. On the basis of the existing literature on the effects of fMRI adaptation [59], we were cautious in making clear-cut predictions about the direction of the effect, i.e., whether facilitation or interference should be observed following adaptation. Therefore, we favoured a bidirectional hypothesis;

- hypothesis 2 (HP2): based on the distinction between mechanisms for arm transport and hand preshaping [30, 31], we expected reaching (i.e., arm transport common to grasp and point actions) and grasp preshaping (specific of the grasp action) to independently impact numerical processing. Specifically, we reasoned that numerical processing, hand reach and grasp might specifically share common cognitive mechanisms for locating items in space and for estimating magnitudes. While locating items is necessary to define reach trajectory, estimating magnitude is required to select the appropriate grip aperture during grasping. We hypothesized that the same computations are carried out also during numerical processing to localise numbers and estimate numerical magnitudes in a mental number space. In line with this view, a previous study by Wiemers et al. [67] has shown that associations between number and physical magnitudes occur independently from associations between number and spatial locations, indicating that these commonly observed effects can have different origins. We hypothesized that adaptation to grasp preshaping (to grasp vs. other action) or to reaching (to grasp and to point vs. others) would independently modulate the effects of numerical magnitude (small vs. large) and numerical distance (close vs. far numbers) in number comparison. Magnitude and distance effects are classic empirical phenomena tapping the semantic processing of numbers [68]. More precisely, we predicted that adaptation to reaching (i.e., arm transport common to grasp and point actions) could bias the distance effect, because of a common mechanism of spatial localisation, while adaptation to grasp preshaping (specific of the grasp action) could bias the magnitude effect, because of a common mechanism for size estimation [e.g., 69]. Note that these specific predictions stem from our hypothesis of shared common cognitive mechanisms for locating items in space and for estimating magnitudes, and are not directly derived from previous studies, where interactions between number magnitude or distance and hand grasp [1, 49] or hand reach have been observed and discussed more in general [32–36] (see also [65]). Considering the explorative nature of this hypothesis, the direction of our predicted effects was bidirectional.

# Method

## Preregistration details

This study research plan was preregistered on OSF. Details on preregistration can be found here. As of the date of submission of this research plan for preregistration, the data had not yet been collected, created, or realized.

## Sample size and participants

We planned to test a minimum of 24 participants. We used the software program G*Power 3.1.9.4 to define this minimum sample size (planned statistical analysis: ANOVA-repeated measures; power = .99; effect size = .25; alpha error probability = .001; see the OSF registration associated to this study for further details). We planned to use Bayesian methods for data analysis, which has two major advantages. First, Bayesian methods allow to measure evidence for both the alternative and the null hypotheses. Second, as a consequence of testing of both hypotheses, these methods allow to systematically add observations until substantial evidence for one of the two hypotheses is observed. Therefore, we planned to conduct the planned analyses once data from 24 participants would be collected, and to continue data collection until we would have reached enough evidence for the alternative or the null hypothesis. The stopping rule was reached with data from 24 participants. The sample was composed by 24 Italian speakers healthy adults (mean age = 23ys, SD = 4; 16 females) with normal or correct to normal vision. Handedness was both based on self-reports and measured by use of the Edinburgh Handedness Inventory [70]. All participants were right-handed (mean laterality score = 79, SD = 14; [71]). Considering that a number of studies have disclosed the impact of hand counting direction on numerical processing [3, 72], we asked participants to count from 1 to 10 on their hands. The majority of participants started counting with the right-hand (right starters: N = 18). Participants were recruited through advertisements at the University of Padua and by word of mouth, and they received a reimbursement for agreeing to participate. The amount corresponding to the reimbursement was in line with the policy of the Department of General Psychology of the University of Padova. Prior to the beginning of the experimental section, each participant received and signed the informed written consent. The study conformed with the Code of Ethics of the World Medical Associations (Declaration of Helsinki), and it received the approval of the Ethical Committee for the Psychological Research of the University of Padova (Protocol number: 3174).

## Materials and procedure

The experiment consisted of a within-subject design with 3 factors: Action Type (Grasp; Point; Tap; Passive view), Number Magnitude (small numbers: 1–4; large numbers: 6–9), Number Distance (close to the reference: 3–4,6–7; far to the reference: 1–2, 8–9). The experimental session was organised in 12 blocks. At the beginning of each block participants were required to perform a washout task which consisted in stretching their arms and hands for 20 seconds in order to reset the hand motor system. This reset task was inspired by previous studies on motor adaptation (or adaptation of motor-related processes) adopting paradigms including ad-hoc washout phases [73, 74]. Then, the participant performed one condition of the motor task. The required action was indicated on the screen. Specifically, the participant was required to grasp or point to an object for 16 times, to continue tapping on the table for 30 seconds, or to observe an object for 30 seconds, prior to 16 trials of number comparison. The object was a wooden cube varying in size (4 cm$^3$ or 5 cm$^3$) to avoid adaptation to object size across blocks. Cube size was not taken into account in the analyses. Each action within each block was

repeated with self-paced timing. The starting hand position consisted always in maintaining the right mouse button pressed with both the index and the thumb. The mouse was fixed on the table. After 500ms of continued pressing, the participant was presented with a beep sound indicating to begin the required motor action (Fig 1, panel d). In the Grasp condition the participant was instructed to repeatedly grasp and lift the object, each time bringing the hand back to the starting position and pressing the right button before initiating the next grasping. In the Point condition the participant was instructed to repeatedly point the centre of the cube front face, each time bringing the hand back to the starting position and pressing the right button before initiating the next pointing. To prevent participants from counting the number of action repetitions, participants were not aware of the fact that they had to stop after 16 repetitions in the grasp and point conditions and a beep sound indicated to stop the required motor action. In the Tap condition the participant repeatedly tapped the hand on the table, bringing the hand back to the starting position only when hearing the beep sound indicating to end the action, sound which was presented after 30s. The participants tapped the hand on the table at their own pace, but in the instructions the experimenter emphasized to tap neither too slowly nor too quickly. Participants were encouraged to avoid huge movements. Differently from the pointing and the grasping conditions, during tapping the participant's forearm and hand laid on the table, only moving gently up and down, and the arm did not move. The forearm could move just a little to accompany the hand. The movement trajectory (up and down) was minimal as compared to point and grasp. Also, tapping was not directed toward a specific target position, differently from point and grasp. No object was presented during tapping. In the Passive view condition, the participant was instructed to stop pressing the right mouse button as soon as the starting beep sound was heard, and to remain with the hand in the same position close to the mouse while passively viewing the object for 30s, until the end beep sound. During the hand motor task, the other hand lay behind the table with the palm facing down, open and relaxed, on the participant's leg. The PC monitor was placed at a distance of about 65cm from the participant, and it was embedded within a structure made of black paper panels. The Eprime response box, the headphones, the cables, and the microphone on its handmade stand were hidden behind these black panels (Fig 1, panel b). The cube was placed at a distance of approximately 35cm from the hand starting position, centred around the participant's body midline. At the end of the action, the participant was instructed to place the right hand above the leg under the table, while avoiding any other type of movement. The experimenter promptly removed the object from the table and provided the participant with the microphone.

Then, the number comparison task started. In number comparison, participants were presented with a number digit centrally presented on a computer screen in front of them, and they were required to judge whether the digit was larger or smaller than 5 (the set of targets consisted of all digits except 5, which was used as reference). In each trial, a fixation cross (Calibri Light 40, 500ms) was followed by a blank screen lasting for 500ms (not shown in the Figure), and then by a target number (Calibri Light 40) lasting on the screen until verbal response (time limit: 5000ms). After the verbal response, an empty screen appeared (Fig 1, panel e). The experimenter encoded the response manually. The encoding of the response was followed by a 1000ms delay before the beginning of the next trial. Verbal responses were acquired with a microphone connected to a voice key; to prevent measurement errors due to its sensitivity, participants were trained to answer BA/BE for small/large digits (for similar procedures, see: [53–56, 75]). Response mapping was counterbalanced between participants. The experimental setting in the numerical task was similar to the setting during the hand motor task, except for the following: both hands lay under the table, with the palms facing down, open and relaxed, on the participant's legs; the cube was hidden behind the black panels; the

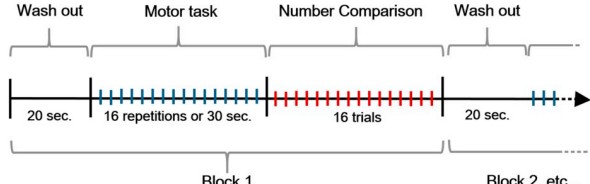

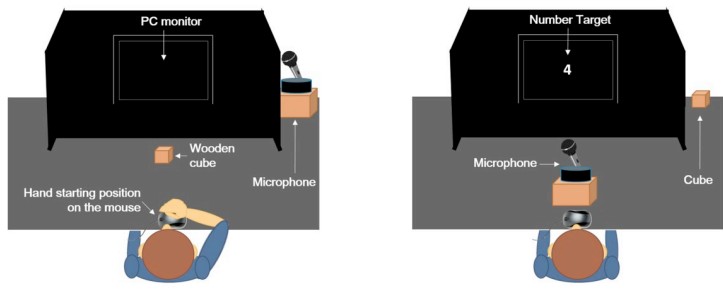

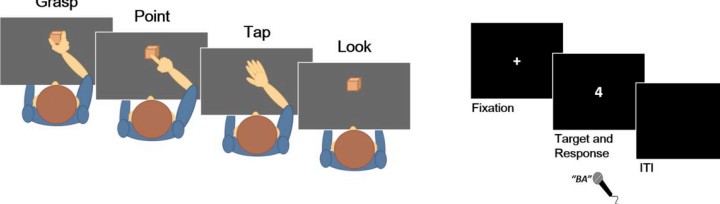

**Fig 1. Graphical illustration of the experimental setup, see main text for details.** Panel a. Timeline of the experiment. Panel b. Experimental setting during the hand motor task. Panel c. Experimental setting during number comparison. Panel d. Illustration of the four conditions of the motor task. Panel e. Schematic illustration of the trial structure in number comparison.

microphone was placed in front of the participant's face, close to the mouth (Fig 1, panel c). Target numbers were presented on the PC monitor during the number task. The participant gave verbal responses on the microphone minimizing as much as possible body movements. Each participant performed a total of 192 numerical trials (4 Action Type * 8 Numbers * 6 repetitions = 192 = 12 trials per condition). The four actions and the eight numerical stimuli were presented in a random order. Consecutive trials with the same number, and consecutive blocks presenting the same action, occurred rarely (2% number repetition, and 2% action repetition). The experiment was built in EPrime (Psychology Software Tools, Pittsburgh, PA). Instructions were presented on the screen at the beginning of the session, and additionally verbally explained by the experimenter. Participants performed a short training before the experimental trials, consisting of a block of motor actions (that included all the motor actions required), and a block of number comparison trials. The timeline of the experiment (Fig 1, panel a), tasks and conditions, and the experimental setting are illustrated in Fig 1. In the same experimental session, each participant performed additional tasks, with the idea to explore interactions between different tasks and factors (i.e., digit naming, affordance judgments, and a manual task in which the participants were required to put in a box eight small objects spread on the

table). Details of these additional tasks can be found in the OSF project associated to this manuscript. Interactions between different tasks and factors are outside the scope of this paper.

## Preregistered analyses

Our dependent variable was the response times (RTs) on correct responses in the numerical task. Trials affected by recording errors (i.e., RTs<120ms or >5000ms, and recording errors detected online by the experimenter) were excluded from the analyses (M = 2%; range: 0%-5%; SEM = 0.3%). Then, the participants' errors in the number task (M = 1%; range: 0–4%; SEM = 0.3%), as well as RTs over 3SD from a participant's mean in correct trials (M = 1%; range: 0%-3%; SEM = 0.1%), were also excluded from the analyses. We analysed the mean RTs by means of a Bayesian repeated measures ANOVA with Type of Action (Grasp, Point, Tap, Passive View), Number Magnitude (small, large), Numerical Distance (close, far) as factors. Bayesian t-tests were used for planned comparisons (Cauchy prior width was set to .707, i.e., the default value set in the JASP software). We considered a Bayes Factor (BF) > 3 to be enough evidence for the alternative hypotheses (HP1 and HP2), and a BF < 1/3 to be enough evidence for the null hypothesis (HP0). Specifically, for testing HP1 (reach and grasp action impacts numerical processing), we considered evidence for the alternative hypothesis a BF > 3 for the factor Action (BF inclusion) in the ANOVA, followed by a BF > 3 for the comparison between RTs for the reach and/or grasp action conditions vs. RTs for the tapping and/or no action conditions. For testing hypothesis 2 (i.e., interaction between Action type, Number Magnitude, and/or Number Distance), we will consider evidence for the alternative hypothesis a BF > 3 for the interaction between Action and one or two numerical variables (BF inclusion), followed by a BF > 3 for the following planned comparisons: the strength of distance effect in the experimental conditions (i.e., Grasp and Point) vs. the amount of distance effect in the control conditions (Tap and Passive View); the magnitude effect in the Grasp condition vs. magnitude effect in the other conditions. We also planned to exclude data from participants not completing the task or being unable to correctly execute the hand actions, but these cases did not occur.

## Exploratory analyses and robustness check

To understand in detail the effects of hand action repetition on the processing of numerical magnitude and distance, we performed a fine-grained analysis of the distance effect through linear regressions. Similar methods were successfully used by previous studies [76–78]. Linear regressions were computed on RTs as a function of the exact number distance of stimuli (i.e., distance from 1 to 4), permitting to estimate at the individual level the distance effect more accurately, since all distances are taken into account. The linear regression slope reveals the size of the distance effect, with more negative values indicating a large distance effect. Taking into account each distance has permitted to reveal cognitive effects which would have been hidden when collapsing distances into close and far levels [78]. Therefore, this analysis might disclose fine-grained interactions between numerical factors and motor action (this point is discussed in the Method section). The slopes in the different experimental conditions were then compared by means of Bayesian t-test. We then confirmed our findings, by showing that the same pattern of results are obtained also by analysing the data with a frequentist approach (ANOVA and t-test).

## Results

### Bayesian ANOVA and planned comparisons

Mean RTs and SEM for each condition are reported in Table 1, and a summary of the results from the preregistered analyses is reported in Table 2. We performed the preregistered

**Table 1. Mean RTs and SEM for each condition.**

| Action Type | Number Magnitude | Number Distance | M (ms) | SEM |
|---|---|---|---|---|
| GRASP | Small | Close | 616 | 19 |
| | | Far | 583 | 18 |
| | Large | Close | 613 | 19 |
| | | Far | 592 | 17 |
| POINT | Small | Close | 619 | 19 |
| | | Far | 612 | 21 |
| | Large | Close | 629 | 23 |
| | | Far | 601 | 17 |
| TAP | Small | Close | 609 | 15 |
| | | Far | 579 | 17 |
| | Large | Close | 616 | 17 |
| | | Far | 598 | 19 |
| PASSIVE VIEW | Small | Close | 619 | 16 |
| | | Far | 588 | 15 |
| | Large | Close | 616 | 20 |
| | | Far | 608 | 19 |

Bayesian ANOVA on mean RTs in number comparison including Action Type (Grasp, Point, Tap, Passive View), Numerical Distance (Close, Far), and Numerical Magnitude (Small, Large) as factors. The best model included Action Type and Distance as main factors with no interactions (BF$_{10}$ = 7.2e+9; BF$_{incl}$ for Action Type = 3.7, BF$_{incl}$ for Distance = 7.5e+8, all BF$_{incl}$ for interactions < .33).

The effect of numerical distance (Close: M = 617, SEM = 18; Far: M = 595, SEM = 18) was coherent with previous studies showing larger RTs for numbers close to the reference as compared to numbers far from the reference in magnitude comparison tasks. Concerning aim 1, we hypothesised a main effect of action on numerical processing: the results showed moderate evidence for the effect of Action Type. Specifically, RTs were slowest following repetitive pointing action (Grasp: M = 601, SEM = 18; Point: M = 615, SEM = 20; Tap: M = 601,

**Table 2. Schematic summary of the results from the repeated-measures Bayesian ANOVA and resulting planned-comparisons.**

| Statistical analysis | Effect | BF |
|---|---|---|
| Repeated-measures Bayesian ANOVA | Action Type | > 3 |
| | Numerical Magnitude | < .33 |
| | Numerical Distance | > 3 |
| | Action Type * Numerical Magnitude | < .33 |
| | Action Type * Numerical Distance | < .33 |
| | Numerical Magnitude * Numerical Distance | < .33 |
| | Action Type * Numerical Magnitude * Numerical Distance | < .33 |
| Planned-comparisons: Bayesian t-test | REACH effect: (Grasp + Point) vs. (Tap + Passive view) | < .33 |
| | GRASP effect: Grasp vs. (Tap + Passive view) | < .33 |
| | POINT vs. the other conditions | > 3 |
| | GRASP vs. the other conditions | < 1 |
| | LOOK vs. the other conditions | < 1 |
| | TAP vs. the other conditions | < 1 |

SEM = 17; Passive view: M = 608, SEM = 17; Fig 2). Concerning aim 2, we hypothesised an effect of action type on the magnitude effect or on the distance effect: the results showed evidence for the null hypothesis. Indeed, preregistered planned comparisons by means of Bayesian t-tests investigating the effects of the *transport component* (reach = Grasp + Point vs. Tap + Passive view) and of *grasp preshaping* (Grasp vs. Tap + Passive view) revealed evidence for HP0 (both $BF_{10} < .33$), suggesting that the effect of Action Type is driven by mechanisms which are different from what expected.

Additional planned comparisons confirmed that the main effect of Action Type was driven by slower responses in number comparison after pointing as compared to the ensemble of the other conditions ($BF_{10} > 6$), while there was no evidence for differences between each other condition and the ensemble of the remaining conditions (all $BF_{10} < 1$). The effect of Action Type is illustrated in Fig 2.

### Fine-grained analysis of the distance effect

While indicating that the hand pointing had a specific effect on performance in number comparison, the results, however, were not completely in line with the hypotheses, because there was evidence for no effect of arm transport or grasp preshaping with respect to the control conditions. To deeper explore the effects of hand actions in numerical processing, we adopted a method which has been found useful in previous studies, consisting in investigating the distance effect by means of the regression analysis for repeated measures [76–78]. One advantage of this method is that all the distance levels can be taken into account, resulting in a more accurate estimation of the distance effect. Therefore, this measure might disclose fine-grained interactions between numerical factors and motor action (this point is discussed in the Method section). The linear regression of the mean RTs as a function of the numerical distance (distances 1–4) was computed for small and large numbers separately. In this way, the regression slope is an index of the distance effect, with a steeper negative slope corresponding to a

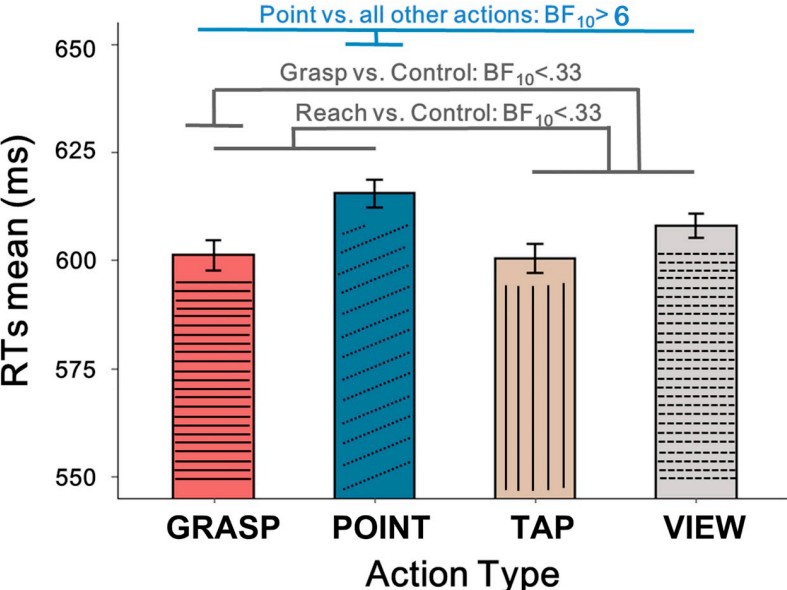

**Fig 2. Effects of action type.** Preregistered planned comparisons and the related results are reported in grey (grasp vs. control conditions; reach vs. control conditions), while additional planned comparisons and the related results are reported in blue. Error bars represent within-subjects SEM, computed only for illustration purposes [79].

stronger distance effect. RTs for each condition are illustrated in Fig 3a, and the corresponding slopes are illustrated in Fig 3b. Interestingly, we found that on average the distance effect for small numbers increased (i.e., it was more negative) after grasping as compared to the ensemble of the other conditions (t(23) = -2.8, p = .01, $BF_{10}$ >4), while it was reduced (i.e. it was closer to 0) after pointing as compared to the ensemble of the other conditions (t(23) = -3.1, p = .006, $BF_{10}$ >7). The distance effect after passive view or tap did not differ from the ensemble of the other conditions (all p>.1, all $BF_{10}$ < .33). The same set of analyses applied to slopes for

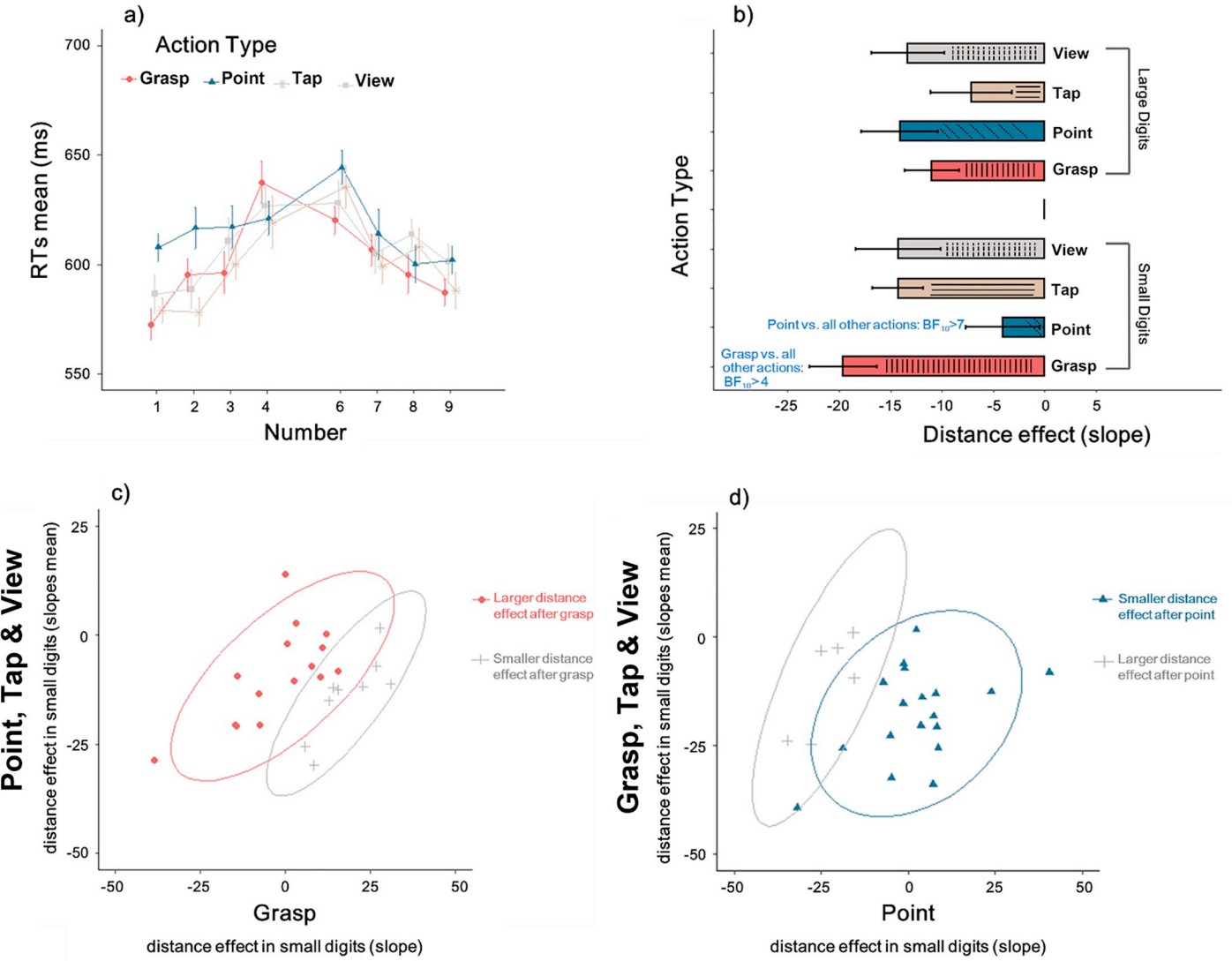

**Fig 3. Panel a. RTs as a function of number target and action type**. Error bars represent within-subjects SEM, computed only for illustration purposes [74]. **Panel b. Regression slopes representing the size of the distance effect for small or large digits in each action type condition**. Error bars represent SEM. Results from exploratory analyses are written in blue. **Panel c. Individual regression slopes as a function of numerical distance in the grasp–small number condition**. Negative slopes indicate performance in the direction of the classical distance effect (i.e., faster responses for digits far from the reference as compared to closer ones). Red circles indicate more negative slopes for small numbers after grasping as compared to the other conditions. Grey crosses indicate more negative slopes for small numbers after tapping, pointing, and viewing as compared to the grasping condition. **Panel d. Individual regression slopes as a function of numerical distance in the point–small number condition**. Negative slopes indicate performance in the direction of the classical distance effect (i.e., faster responses for digits far from the reference as compared to closer ones). Blue triangles indicate less negative slopes for small numbers after pointing as compared to the other conditions. Grey crosses indicate less negative slopes for small numbers after tapping, grasping, and viewing as compared to the pointing condition.

large numbers did not reveal differences among conditions (all p>.1, all $BF_{10} < .5$). More negative slopes for small digits after grasp action as compared to the other conditions were found in the two thirds of the participants (63%: 15/24; Fig 3c), while less negative slopes for small digits after point action as compared to the other conditions were found in the three fourths of the participants (75%: 18/24; Fig 3d). When comparing the distance effect for small numbers after grasping or pointing with the other conditions separately, only the comparison between grasping and pointing supported the alternative hypothesis (BF = 14). The other comparisons (grasping or pointing vs. tapping or reaching) neither favoured the null nor the alternative hypothesis (BF range: .45–2.4).

## Robustness check

For completeness, and to validate our results, we also run the analyses using a frequentist approach. The results from an ANOVA with Action Type, Distance, and Magnitude as within/subjects factors confirmed the significant main effects of Action Type ($F(3,69) = 3.5$, p = .02, $\eta_p^2 = .133$) and Distance ($F(1,23) = 48.4$, p < .0001, $\eta_p^2 = .678$), and revealed a significant interaction between all factors ($F(3,69) = 3.5$, p = .02, $\eta_p^2 = .13$). The other main effects or interactions were not significant (all p>.1). Planned comparisons did not find effects of the *transport component* (reach = Grasp + Point vs. Tap + Passive view, p >.1) or of *grasp preshaping* (Grasp vs. Tap + Passive view, p >.1). Additional planned comparisons confirmed that the main effect of Action Type was due to slower responses in number comparison after pointing (point vs. all other conditions: t(23) = 2.9, p = .008, Cohen's d = .6). The comparison of each other condition against the ensemble of the remaining conditions did not reveal significant differences (all p >.05). Finally, the results from the exploratory analyses on the distance effect also confirmed the findings from the Bayesian approach: the distance effect for small numbers increased after grasping (t(23) = -2.8, p = .01, Cohen's d = -.572) and decreased after pointing (t(23) = -3.1, p = .006, Cohen's d = .622) as compared to the ensemble of the other conditions, while there was no significant difference for the other comparisons (all p>.1). The independent comparison between the distance effect for small numbers after grasping vs. the distance effect for small numbers after pointing was also significant (t(23) = -3.4, p = .003, Cohen's d = -.687).

## General discussion

In this study we investigated the effects of repetitive hand action on the performance in a subsequent number comparison task. The participants performed pointing, grasping, tapping or passively viewing at actions in separate blocks. Following each action phase, they were presented with a number digit on the screen, and they orally indicated whether the number was smaller or larger than five. We predicted effects of arm transport (taking place both in pointing and grasping) or grasp preshaping (present only in grasping) on the main performance in number comparison, as well as on the processing of numerical distance and/or magnitude. A series of preregistered and exploratory analyses were conducted. While arm transport and grasp preshaping do not significantly affect numerical processing, we found a reduced distance effect for small numbers after pointing, while it was enhanced by grasping.

The main novel finding is that RTs in number comparison were significantly slower after repetition of pointing as compared to any other action. A unique effect of pointing is different from what we had predicted based on the combination of results from studies on human grasping and studies on numerical processing. Indeed, studies on human grasping make a clear distinction between arm transport and grasp preshaping [29], and highlight different cognitive and neural mechanisms associated to these two components of hand movements [30, 31]. In

many studies the action of pointing is often considered as representative of reaching, i.e. of arm transport (e.g., [80, 81]; for a discussion on this topic, see: [31]). In fact, pointing is not exactly an equivalent of reaching, because pointing can be executed both with and without arm transport. Indeed, in a more recent study [31] the authors highlight a series of distinct neural activations for pointing as compared to arm transport, and they underline the communicative nature of pointing, suggesting that the uniqueness of this action might arise from a combination of visuospatial processing for object location and processing related to social cognition [82, 83].

In child development, starting from one year old, index finger pointing has an important social role, since it conveys interactions with the caregiver [84]. More in general, pointing is part of gesturing during speech in many different contexts. Importantly, researchers have observed that children tend to learn maths concepts such as mathematical equivalence better when teachers encourage them to associate the correct pointing gesture to a given problem [85]. It appears clear that the embodiment of pointing, through its repetitive use in gesturing, should constitute a scaffold for the development of number knowledge (for related concepts, see: [86]). Importantly, pointing is particularly relevant in the context of numerical cognition also because it is the action through which children learn counting arrays of visual items [87]. In this sense, the routine of index finger counting might elicit the understanding of ordinality (e.g., *I live on the third floor*) before the understanding of cardinality (e.g., *I see three puppies*), suggesting that the former is a prerequisite of the latter [88, 89].

Many studies have underlined that numerical order is processed by mechanisms that are at least partially separated from the ones involved in the processing of number magnitude [90]. One example comes from studies observing the distance effect during comparison of non-numerical ordered sequences [91]. Furthermore, it has been observed that judging order or magnitude of pairs of digits can differently interact with the numerical distance [92]. This suggests that, depending on the task, different processes—possibly serial search and magnitude comparison—are preferentially recruited (see also [93]). In addition, other studies indicated that order can be automatically activated during number magnitude comparison [94], albeit the distance effects in magnitude and order tasks are not correlated [95]. Finally, some studies have argued that the automatic processing of number order may explain biases in magnitude tasks, biases typically attributed to the unique effect of magnitude (e.g. [96, 97]; see also: [65]). In view of these studies, we propose that pointing might trigger mechanisms implied in the processing of order information due to its link to actions of ordering and counting, consequently impacting number magnitude comparison. Following this reasoning, we should expect to observe a facilitation effect of pointing on tasks requiring the processing of numerical order. Future studies are necessary to test this prediction.

We could not find a link between number magnitude and grasping, and between number distance and reaching, as originally predicted (hypothesis 2). Nonetheless, the exploratory analyses highlighted differential fine-grained effects of grasping and pointing on numerical distance. Specifically, statistical analyses indicated that, for small numbers only, the distance effect is exacerbated by immediate repetition of grasping action, while, in contrast, it disappears when number comparison is preceded by repetitive pointing. This effect was not predicted and it requires a follow-up study in order to be replicated and interpreted. We advance here one possible interpretation which should be tested in future studies: considering that the origin of the distance effect is rooted in number semantics [68], these results might reflect shared mechanisms between grasp and number cardinality, on the one hand, and between point and ordinality, on the other hand (see also [65]). Specifically, participants were more sensitive to numerical distance after grasping because the task required to process number magnitude, i.e., number cardinality. On the contrary, pointing reduced the sensitivity to numerical distance.

It is worth mentioning that many studies have investigated the differences in the processing of number cardinality and ordinality [90]. Wiemers et al. [67] suggested that number-size congruency and number-space congruency effects might be related to different aspects of numerical processing, and specifically to the processing of number cardinality and ordinality, respectively. In agreement with this claim, here we further suggest that grasping shares with number cardinality the need to process size (object size or number size), while pointing might share with number processing mechanisms related to the ordering and counting of items. With regard to the effect of grasping, the ATOM theory suggests that physical and numerical magnitudes are commonly represented, and this magnitude system is devoted to action, thus providing an account for the interactions between numerical magnitude and grasping. In contrast, the MNL account suggests that numbers are spatially represented along a continuum, and it has been shown that this spatial format of representation is also used for other types of ordinal sequences, as well as for sequences of numbers stored in working memory, independently on their magnitude [e.g., 97]. Therefore, the MNL account might also explain interactions between numerical processing and pointing.

Interestingly, the grasping and pointing influenced distance effect only for small numbers (<5), while for larger number (>5) the distance effect was not modulated by the type of prior action. In line with this finding, previous studies had already noticed that some behavioural effects ascribed to the functional relation between grasping and number magnitude suggest a specific link between precision grip and small numbers [49–53]. These observations suggest that the mental representation of small numbers might be particularly sensitive to embodiment, possibly because, in everyday life, precision grip is mostly performed by one hand, as well as counting of small quantities [53].

That said, future studies are needed to confirm and clarify the exact mechanisms relating grasping and pointing to number magnitude and distance. In particular, future studies should also consider the use of different control conditions. In fact, while controlling for object-related processes, arm transport, and hand pre-shaping, both observation and tapping are not free from sensorimotor processing, with observation activating object affordances, and tapping being subserved by sensorimotor processing (processing which is however different from the one involved in grasping and pointing). In this sense, different or additional control conditions will help to clarify the results of this study. Another interesting point to consider in future studies concerns the factors contributing to the variability in the observed effects. The results of our fine-grained analysis showed that not all the participants' behaviour was in the direction revealed by the group analyses (see also Fig 3, panels c-d). A recent study by Cipora et al. [98] has highlighted the importance of investigating the reliability of psychological effects at the individual level, also considering that effects observed at the group-level might be driven by the performance of few individuals, and therefore they might not be representative of more general cognitive processing. Also, interindividual differences might be related to neuroanatomical differences among individuals. For instance, Krause et al. [99] showed that grey matter volume in different brain regions correlated with the strength of number-space interactions (small numbers associated to left-sided responses, and vice versa) and number-action interactions (small number associated to soft response, and vice versa) at the individual level. Considering this, it will be important in future studies to clarify the incidence of the effects and the factors contributing to individual differences.

## Conclusion

In this study we observed that the action of pointing impacts the processing of numerical magnitudes. This finding confirms previous studies on interactions between numerical processing

and motor action [49–53, 63, 100], in line with embodied and grounded cognition accounts [10, 11]. More than this, this study gives new insights for future lines of research, suggesting that the meaning of each hand action should be taken into account when studying number-action interactions. Indeed, we suggest that the effects of pointing in number comparison might stem from the functional role of pointing in numerical processing. In his sense, pointing—as well as it was shown for grasping [49]—might have a *special status* in numerical processing, possibly due to its repetitive use in counting and in conveying spatial information during child development. Finally, we have described here the effectiveness of a promising method to study number-action interactions, which is based on the neurophysiological principles of neural adaptation [57] thus permitting to advance convincing claims on the functional link between motor action and numerical processing.

## Acknowledgments

We are grateful to Stefano Massaccesi for his contribution in preparing the experimental setting, and to Jacopo Torre and Diego Varotto for their help in managing the experimental room.

## Author Contributions

**Conceptualization:** Mariagrazia Ranzini, Carlo Semenza, Marco Zorzi, Simone Cutini.

**Data curation:** Mariagrazia Ranzini.

**Formal analysis:** Mariagrazia Ranzini.

**Investigation:** Mariagrazia Ranzini.

**Methodology:** Mariagrazia Ranzini.

**Project administration:** Mariagrazia Ranzini.

**Supervision:** Carlo Semenza, Marco Zorzi, Simone Cutini.

**Visualization:** Mariagrazia Ranzini, Simone Cutini.

**Writing – original draft:** Mariagrazia Ranzini.

**Writing – review & editing:** Mariagrazia Ranzini, Carlo Semenza, Marco Zorzi, Simone Cutini.

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
