## [Decision Letter · Decision Letter 0]

24 Jan 2022

PONE-D-21-36636Influences of hand action on the processing of symbolic numbers: a special role of pointing?PLOS ONE

Dear Dr. Ranzini,

Thank you for submitting your manuscript to PLOS ONE. I have sent it to three expert reviewers and have now received their comments back. As you can see in the comments at the bottom of this email, all three reviewers find merit in the manuscript, reporting an interesting study with sound methods. I concur with this general assessment. However, the reviewers also make several suggestions that I think could help improve the manuscript further. I will not reiterate these comments here as the reviewer are quite clear. But I encourage you to address all of these points in a revised version of the manuscript. Please submit your revised manuscript by Mar 10 2022 11:59PM. If you will need more time than this to complete your revisions, please reply to this message or contact the journal office at plosone@plos.org. Please include the following items when submitting your revised manuscript:A rebuttal letter that responds to each point raised by the academic editor and reviewer(s). You should upload this letter as a separate file labeled 'Response to Reviewers'.A marked-up copy of your manuscript that highlights changes made to the original version. You should upload this as a separate file labeled 'Revised Manuscript with Track Changes'.An unmarked version of your revised paper without tracked changes. You should upload this as a separate file labeled 'Manuscript'.

We look forward to receiving your revised manuscript.

Kind regards,

Jérôme Prado

Academic Editor

PLOS ONE

Journal Requirements:

“This study was carried out within the scope of the research program Dipartimenti di Eccellenza (art.1, commi 314-337 legge 232/2016), which was supported by a grant from MIUR to the Department of General Psychology, University of Padua. Also, this work was funded by the European Union’s Horizon 2020 research and innovation program under Marie Skłodowska-Curie Grant 839394 (to M.R.).”

Reviewers' comments:

Reviewer #1: I have already reviewed this paper in another journal. Consequently, after a careful reading, I have only updated my review from this new submission. In this study, authors used different actions and then assess their links with a classical numerical task. The finding is very interesting and important, and could be published after several minir improvements.

Abstract :

Embodiment is now defined but a reader needs an example (a short one).

Introduction:

We need several references for the sentence: “In line with this idea, an increasing number of behavioural studies have described interactions between symbolic number processing and the hand action of reaching and grasping objects, suggesting overlapping mechanisms between the mental representation of numbers and sensorimotor processes.”

All the following paragraph is not accompanied by a clear explanation on the mental number line; please, we need this additional information. (For instance, rightward shifts in reaching trajectories during pointing tasks were positively correlated with numerical magnitude, so that targets consisting of larger digits biased finger trajectories more rightward as compared to smaller digits (e.g., Song, & Nakayama, 2008; Rugani, Betti, & Sartori, 2018). Similarly, when required to grasp and freely displace an object while performing at the same time a comparison of Arabic digits magnitude, participants systematically misplaced the object leftward while processing smaller digits as compared to larger ones (Gianelli, Ranzini, Marzocchi, Micheli, & Borghi, 2012). Finally, reach velocity while connecting with the index finger numerical stimuli on a touch screen was higher with more distant numbers than closer ones (e.g., 1- 2 vs. 1-8: Girelli et al., 2016).)… for example, after this sentence: “Taken together, these interactions between number and reaching are in line with and support the well-established hypothesis of the mental number line, postulating that the mental representation of numbers is spatial in nature, and oriented from left to right, at least in Western societies as a possible consequence of reading and writing habits (e.g., [37]).” Authors could add something like that: in this view, a cue like a small number can enhance the processing of a reaching task toward the left side of the space, etc….

It is strange to explain that the present studies uses “neural adaptation” and then performed actually a behavioral task without neural investigation… Is there any similar paradigm in the behavioral domain? Motor adaptation? Motor learning? To be clearer, it could be interesting to keep a related paradigm for the present study.

Reviewer #2: The manuscript presents a novel and interesting study investigating the impact of hand action on the processing of symbolic numbers. The authors found slower reaction times in the number comparison task after repetitive pointing. The distance effect, a classical signature of semantic number processing, was diminished after performing pointing and enhanced after performing grasping. The study elegantly implies a classical neural adaptation paradigm, thus temporally dissociating manual actions and number processing. A study was pre-registered. Moreover, Bayesian statistical analysis was used to draw conclusions about both positive and negative findings. Methodologically speaking, this is a powerful and innovative study.

I do not have major concerns regarding the manuscript. However, several terminological clarifications and formatting improvements would be beneficial for the manuscript’s clarity, and several relevant sources could enrich the interpretation of the findings.

Minor concerns:

1. The authors draw a parallel between nature vs. nurture / grounded vs. embodied cognition (line 46). As I understand these terms, nature relates here to inherited properties of the cognitive system, i.e., those that are genetically/biologically predefined. In contrast, grounded representations (i.e., an association between gravity and number/space) are still learned in the course of individual cognitive development. I would either clarify this parallel or remove it from the text as confusing.

2. Throughout the text (e.g., lines 45 vs. 51), the term grounded is used in two meanings: one is like in line 45 (grounded cognition, i.e., constraints of the physical world reflected in cognitive processes); the other one is like in line 51 (“cultural habits… grounded on cognitive and neural constraints”, i.e., having neuronal or cognitive correlates). I would recommend the authors either make an explicit distinction between the two meanings or replace the term with a different one in one type of context.

3. The authors use the term number processes (e.g., line 67), which sounds unclear to me. I am more familiar with number processing or numerical processing. I would stick to these more established terms if this is what is meant. In that case, the expressions like “number processes… are grounded on neurocognitive constraints” (lines 67-68) could be reformulated as “numerical processing is constrained by neuronal mechanisms”, which, in my opinion, is more theoretically precise. See also “process of numerical magnitude and distance” (lines 310-311) for the same problem.

4. The authors review the literature on the SNARC effect (lines 80-94), but only the part that investigated the effect in hand responses. However, spatial-numerical associations were also demonstrated for other effectors (eye movements, Myachykov et al., 2016; foot responses, Schwarz and Müller, 2006; full-body movements, see for reviews Fischer and Shaki, 2014; Toomarian and Hubbard, 2018). Whether numbers are associated with hand or lateralized space, this distinction is known in the literature as the location vs. effector hypothesis. It is discussed, for example, in this publication: Gut, M., Binder, M., Finc, K., & Szeszkowski, W. (2021). Brain activity underlying response induced by SNARC‑congruent and SNARC‑incongruent stimuli. Acta Neurobiol Exp, 81. It is worth mentioning that there is more support in favor of the location hypothesis. I do not insist that the authors refer to this literature in their introduction, but they might want to consider it in the discussion section.

5. The authors discuss pointing (lines 80-94) and grasping (lines 95-107) in numerical tasks; however, one of the references in the former para is about grasping (reference 35).

6. My impression is that the authors unnecessary parallel pointing vs. grasping with MNL vs. ATOM accounts: these two new concepts (MNL and ATOM) are not used in the analysis or interpretation of results and only appear once again in the introduction. I would recommend removing them entirely and speaking instead about ordinal vs. cardinal aspects of number meaning, like in this work (also relevant for the current study): Wiemers, M., Bekkering, H., & Lindemann, O. (2017). Two attributes of number meaning. Experimental psychology.

Otherwise, the authors might want to relate their findings to these two theories (MNL and ATOM) in the discussion section.

7. It is also possible that the two aspects (ordinal vs. cardinal) diverge at the inter-individual level, i.e., some participants rely more on one, while other participants on the other aspect: see Krause, F., Lindemann, O., Toni, I., and Bekkering, H. (2013). Different brains process numbers differently: structural bases of individual differences in spatial and nonspatial number representations. J. Cogn. Neurosci. 26, 768–776. doi: 10.1162/jocn_a_00518

8. Unfortunately, I was not able to understand how exactly the randomization was performed (lines 240-245). In particular, the authors write here: “Consecutive blocks presenting the same action, were highly improbable cases.” – Does this mean that participants could potentially perform the same action block (tapping, pointing, etc.) several times? If not, then what exactly method was used to randomize blocks (e.g., true randomization, Latin square, etc.)?

9. I found it inconvenient to shuttle back and forth between the main text and a long detailed description of Fig. 1. I would recommend shifting all relevant details about the procedure to the main text while keeping the figure caption short.

10. It is written in the capture for Fig. 1: “a minimum of 1000ms and in any case after manual encoding of the participant’s response by the experimenter.” Does this mean that the experimenter initiated each trial, e.g., by a button press? If yes, this should be stated in the manuscript explicitly.

11. Through the text, the authors report SEM, even for descriptive statistics of accuracy (lines 289-291) or RT (Table 1). I am more familiar with reporting SDs in these contexts. Is there a particular reason for reporting SEM instead?

12. The authors introduce the term transport component, an average of grasping and pointing conditions (line 344). But tapping also includes hand movement to the object. It probably requires a better term or a justification for why tapping is not a part of the transport component.

13. It remained unclear to me how the clusters (Fig. 3c and 3d) were defined. Was a cluster analysis applied, or was a particular numerical threshold introduced on one or both variables? Since some observations from different clusters overlap on each variable, I assume the former. If this is the case, the clustering method should be described in detail. These figures are pretty complicated and are not referred to in Discussion. Perhaps, they could even be omitted in the manuscript if they do not add substantial evidence in favor of any of the hypotheses.

14. The authors hypothesize that ordinality is acquired earlier than cardinality (lines 440-443). I would refer here to a model of early mathematical development by Krajewski, which supports this idea. See Fig. 1 at: Krajewski, K., & Schneider, W. (2009). Early development of quantity to number-word linkage as a precursor of mathematical school achievement and mathematical difficulties: Findings from a four-year longitudinal study. Learning and instruction, 19(6), 513-526.

15. In Conclusion, the authors claim that the study supports bidirectionality of action-number links (line 476). However, bidirectionality implies that numbers also influence actions. The authors did not test this direction in their study. I would remove this term.

16. I would recommend the authors summarize once again the main findings of the study in one or two sentences in Conclusion, right after the first sentence (line 477), and before discussing methodological advances and suggestions for future research.

17. I personally found it confusing to see a citation at the end of the conclusion, especially the one that includes “see also”. It makes an impression of an incomplete text. I would omit the last citations entirely, or if it is crucial for the authors to keep them, rephrase the sentence so that the references appear at its beginning.

Reviewer #3: The authors investigated whether repetitive hand action affects number processing. The originality of the study lies in the fact that they studied the interaction between action and numbers with functionally and temporally dissociated tasks. The methods are sound (I only have one concern reported below) and the manuscript is overall well written. I would however like the authors to precise their hypotheses and interpretations and centre the introduction on their research question and what they can tell about their results. More precisely, my concerns/questions are:

-I am not convinced that the first part of the introduction on finger use is relevant for the paper (certainly not in these proportions). I understand that the link between finger use and number processing is more eloquent than the link between action (grasping, pointing) and number processing. However, finger use and actions directed towards objects are fundamentally different and their link with number processing might be too. As said by the authors, the link between numbers and fingers might arise from finger counting habits, while the link between numbers and action might arise from the common need to process magnitude (object size, distance and number magnitude). An alternative might be that number is grounded in hand sensorimotor experience, but I am not convinced that the current results on action allow drawing conclusions on the link between hand representation and number in a more general way (which, by the way, the authors didn’t do). If, however, the authors wanted to go this way, I would advise them to make clearer hypotheses and discuss their results in this framework. Here, for instance, they make the hypothesis that finger tapping should have no effect on subsequent number processing, which seems to me in contradiction with number processing being grounded in hand sensorimotor experience. Otherwise, I would suggest focusing the introduction on the link between number and action to go straight to the point and not lose the reader along the way.

-Most effects of number processing on action (or conversely) are « lateralized »: small/large numbers induce leftwards/rightwards deviations in hand trajectory and increase/decrease hand opening (or conversely). In the present paper, the authors choose to test the effect of action on number processing by using centralised movement without taking into account grip opening. May the authors explain this choice? Could they also discuss how this could explain their results?

-From the papers cited by the authors (e.g., Badets et al., 2010) it is not clear why the authors hypothesized that grasp preshaping and reaching would affect number processing differently. Could they clarify which previous results could predict that grasping, but not reaching, could affect small numbers or number distance (or something like that)? Please also clarify how they could be different (without necessarily specifying the direction), but the expected effects remain a bit obscure to me.

-Please note that Geers et al. (2021) also showed an effect of a functionally unrelated (but, concurrent) action on number comparison.

-In the fine-grain analysis of the distance effect, the difference between grasping and the ensemble of the other conditions seems to be driven by the small distance effect in the pointing condition. Wouldn’t it be more relevant to compare grasping to each of the other conditions? It would guess that only the difference with pointing would be different, supporting a specific effect of pointing.

-Might the authors develop a bit more there how their results support a specific link between grasp and cardinality and point and ordinality?

-Unless I missed it, the link to access the data on OSF has not been provided.

Laurie Geers

6. PLOS authors have the option to publish the peer review history of their article (what does this mean?). If published, this will include your full peer review and any attached files.

Reviewer #1: No

Reviewer #2: **Yes: **Alex Miklashevsky

Reviewer #3: **Yes: **Laurie Geers

---

## [Author Response · Author response to Decision Letter 0]

12 Apr 2022

Response to the Reviewers’ comments

Reviewer #1

Reviewer 1, point 1. I have already reviewed this paper in another journal. Consequently, after a careful reading, I have only updated my review from this new submission. In this study, authors used different actions and then assess their links with a classical numerical task. The finding is very interesting and important, and could be published after several minir improvements.

Reply. We thank Reviewer 1 for the positive evaluation of our work, as well as for insightful comments during the previous submission. We have further considered all the reviewer’s comments, as indicated here below.

Reviewer 1, point 2. Abstract: Embodiment is now defined but a reader needs an example (a short one).

Reply. We have rephrased the beginning of the abstract to provide the reader with a short example related to the present study (page 1, lines 5-6).

Reviewer 1, point 3. Introduction: We need several references for the sentence: “In line with this idea, an increasing number of behavioural studies have described interactions between symbolic number processing and the hand action of reaching and grasping objects, suggesting overlapping mechanisms between the mental representation of numbers and sensorimotor processes.”

Reply. Relevant references have been added to the sentence at issue (page 3, line 52).

Reviewer 1, point 4. All the following paragraph is not accompanied by a clear explanation on the mental number line; please, we need this additional information. (For instance, rightward shifts in reaching trajectories during pointing tasks were positively correlated with numerical magnitude, so that targets consisting of larger digits biased finger trajectories more rightward as compared to smaller digits (e.g., Song, & Nakayama, 2008; Rugani, Betti, & Sartori, 2018). Similarly, when required to grasp and freely displace an object while performing at the same time a comparison of Arabic digits magnitude, participants systematically misplaced the object leftward while processing smaller digits as compared to larger ones (Gianelli, Ranzini, Marzocchi, Micheli, & Borghi, 2012). Finally, reach velocity while connecting with the index finger numerical stimuli on a touch screen was higher with more distant numbers than closer ones (e.g., 1- 2 vs. 1-8: Girelli et al., 2016).)… for example, after this sentence: “Taken together, these interactions between number and reaching are in line with and support the well-established hypothesis of the mental number line, postulating that the mental representation of numbers is spatial in nature, and oriented from left to right, at least in Western societies as a possible consequence of reading and writing habits (e.g., [37]).” Authors could add something like that: in this view, a cue like a small number can enhance the processing of a reaching task toward the left side of the space, etc….

Reply. We agree with the reviewer that our explanation of the MNL was not exhaustive and could not be clear enough for a reader with no experience in the field of numerical cognition. Thanks to this comment, we have now revised the sentence following the reviewer suggestion (page 4, lines 76-81; see also our reply to Reviewer 2’s point 4).

Reviewer 1, point 5. It is strange to explain that the present study uses “neural adaptation” and then performed actually a behavioral task without neural investigation… Is there any similar paradigm in the behavioral domain? Motor adaptation? Motor learning? To be clearer, it could be interesting to keep a related paradigm for the present study.

Reply. We understand the point of view of the reviewer. As far as we know, there is no systematic literature investigating the effects of motor adaptation or motor learning in number processing, albeit the literature on motor learning and motor adaptation in other fields of experimental psychology is surely very rich. Considering that the original rationale of this study was based on the idea to induce adaptation at the neural level, we prefer to avoid to completely rewrite our rationale in order to reframe it on a different literature. However, we have rephrased and reorganised some parts of the manuscript (abstract, and pages 5-6, lines 108-121) to further clarify the idea that motor adaptation should induce neural adaptation, and we have also described some studies on numerical processing adopting similar approaches.

Reviewer #2

Reviewer 2, point 0. The manuscript presents a novel and interesting study investigating the impact of hand action on the processing of symbolic numbers. The authors found slower reaction times in the number comparison task after repetitive pointing. The distance effect, a classical signature of semantic number processing, was diminished after performing pointing and enhanced after performing grasping. The study elegantly implies a classical neural adaptation paradigm, thus temporally dissociating manual actions and number processing. A study was pre-registered. Moreover, Bayesian statistical analysis was used to draw conclusions about both positive and negative findings. Methodologically speaking, this is a powerful and innovative study.

I do not have major concerns regarding the manuscript. However, several terminological clarifications and formatting improvements would be beneficial for the manuscript’s clarity, and several relevant sources could enrich the interpretation of the findings.

Reply. We sincerely thank Reviewer 2 for the positive evaluation of our work. We have considered all the comments and suggestions, as reported here below.

Reviewer 2 point 1. Minor concerns: 1. The authors draw a parallel between nature vs. nurture / grounded vs. embodied cognition (line 46). As I understand these terms, nature relates here to inherited properties of the cognitive system, i.e., those that are genetically/biologically predefined. In contrast, grounded representations (i.e., an association between gravity and number/space) are still learned in the course of individual cognitive development. I would either clarify this parallel or remove it from the text as confusing.

Reply. Also following Reviewer 3’s comment, we have opted for removing this parallelism. We have also modified the sentences where the word “grounded” had a non-conventional meaning. 

Reviewer 2, point 2. 2. Throughout the text (e.g., lines 45 vs. 51), the term grounded is used in two meanings: one is like in line 45 (grounded cognition, i.e., constraints of the physical world reflected in cognitive processes); the other one is like in line 51 (“cultural habits… grounded on cognitive and neural constraints”, i.e., having neuronal or cognitive correlates). I would recommend the authors either make an explicit distinction between the two meanings or replace the term with a different one in one type of context.

Reply. As also mentioned in our previous reply, we have modified the previous sentences (now lines 36-40) to avoid using the term “grounded” with different meanings.

Reviewer 2, point 3. 3. The authors use the term number processes (e.g., line 67), which sounds unclear to me. I am more familiar with number processing or numerical processing. I would stick to these more established terms if this is what is meant. In that case, the expressions like “number processes… are grounded on neurocognitive constraints” (lines 67-68) could be reformulated as “numerical processing is constrained by neuronal mechanisms”, which, in my opinion, is more theoretically precise. See also “process of numerical magnitude and distance” (lines 310-311) for the same problem.

Reply. As suggested, we have changed from the use of the terms “number processes” to the use of the terms “numerical processing” or “symbolic number processing”. We have also changed the terms used in lines 67-68 and 310-311 (now lines 48-49 and 331-332), accordingly to the reviewer’s suggestions.

Reviewer 2, point 4. 4. The authors review the literature on the SNARC effect (lines 80-94), but only the part that investigated the effect in hand responses. However, spatial-numerical associations were also demonstrated for other effectors (eye movements, Myachykov et al., 2016; foot responses, Schwarz and Müller, 2006; full-body movements, see for reviews Fischer and Shaki, 2014; Toomarian and Hubbard, 2018). Whether numbers are associated with hand or lateralized space, this distinction is known in the literature as the location vs. effector hypothesis. It is discussed, for example, in this publication: Gut, M., Binder, M., Finc, K., & Szeszkowski, W. (2021). Brain activity underlying response induced by SNARC‑congruent and SNARC‑incongruent stimuli. Acta Neurobiol Exp, 81. It is worth mentioning that there is more support in favor of the location hypothesis. I do not insist that the authors refer to this literature in their introduction, but they might want to consider it in the discussion section.

Reply. We thank the reviewer for this important comment. We had focused on SNARC-like studies investigating the effect in hand responses because we reasoned that these studies were more in line with the aim of our experiment. That said, we agree that showing SNARC with hand responses does not mean that number-space associations are effector specific. The location hypothesis in our view is more in agreement with the idea that there are common mechanisms between hand action and numerical processing (such as mechanisms for locating items in the external space and numbers in the internal mental space). We have briefly clarified our view by also referring to some of the studies suggested by the reviewer (page 4, lines 78-81).

Reviewer 2, point 5. 5. The authors discuss pointing (lines 80-94) and grasping (lines 95-107) in numerical tasks; however, one of the references in the former para is about grasping (reference 35).

Reply. In fact, we did not discuss pointing and grasping in numerical tasks, instead we discussed reaching and grasping. Reference 35 is about grasping, but the observed effect concerns reaching (final position of the object, so the effect is related to arm transport, which is commonly involved both in pointing and in grasping movements). We have now clarified this issue by specifying that the task of reference 35 was to reach, grasp, and displace the object (page 4, line 69).

Reviewer 2, point 6. 6. My impression is that the authors unnecessary parallel pointing vs. grasping with MNL vs. ATOM accounts: these two new concepts (MNL and ATOM) are not used in the analysis or interpretation of results and only appear once again in the introduction. I would recommend removing them entirely and speaking instead about ordinal vs. cardinal aspects of number meaning, like in this work (also relevant for the current study): Wiemers, M., Bekkering, H., & Lindemann, O. (2017). Two attributes of number meaning. Experimental psychology.

Otherwise, the authors might want to relate their findings to these two theories (MNL and ATOM) in the discussion section.

Reply. We prefer to keep a brief description of the two accounts in the introduction because in our view they are theoretically related to the effects on numerical processing observed during reaching and grasping tasks. Also, a description of these accounts was recommended by previous reviewers. That said, we completely agree with the reviewer that we should relate our findings to the two theories in the discussion section (pages 23-24, lines 498-510). Also, we thank the reviewer for suggesting reading the paper by Wiemers et al. which seems very related to our study: we have reported this study both in the introduction and in the discussion sections (see also our reply to reviewer 3’s point 3).

Reviewer 2, point 7. 7. It is also possible that the two aspects (ordinal vs. cardinal) diverge at the inter-individual level, i.e., some participants rely more on one, while other participants on the other aspect: see Krause, F., Lindemann, O., Toni, I., and Bekkering, H. (2013). Different brains process numbers differently: structural bases of individual differences in spatial and nonspatial number representations. J. Cogn. Neurosci. 26, 768–776. doi: 10.1162/jocn_a_00518

Reply. We thank the reviewer for this important comment. We completely agree with the view that the two aspects should be investigated considering individual differences. We have added a comment on this idea in the discussion section (lines 524-536), and we are considering to systematically investigate this issue in future studies (see also our reply to the Reviewer 2’s point 13).

Reviewer 2, point 8. 8. Unfortunately, I was not able to understand how exactly the randomization was performed (lines 240-245). In particular, the authors write here: “Consecutive blocks presenting the same action, were highly improbable cases.” – Does this mean that participants could potentially perform the same action block (tapping, pointing, etc.) several times? If not, then what exactly method was used to randomize blocks (e.g., true randomization, Latin square, etc.)?

Reply. As the reviewer has noticed, we were not clear in the description of our blocks randomisation procedure. We have now simplified this point, and provide the reader with additional information (page 13, lines 289-291). Concerning blocks randomisation, a list for the four actions was created in Eprime. Within the list, each action was selected randomly, but instead of assigning a weight of three to each action within the list (which would have produced a full randomisation of the twelve blocks), we set the program in a way that the list should be completed (each action selected once) before being repeated. Therefore, yes, there was 1/3 probability of repeating the same action at block 5 and block 9. This happened once for six participants, and never twice. That said, we have assumed that repetition – and in general the order in which the actions were executed - did not have a specific impact on performance given that there was a wash out period after each action.

Reviewer 2, point 9. 9. I found it inconvenient to shuttle back and forth between the main text and a long detailed description of Fig. 1. I would recommend shifting all relevant details about the procedure to the main text while keeping the figure caption short.

Reply. We have shifted all relevant details from Figure 1 caption to the main text.

Reviewer 2, point 10. 10. It is written in the capture for Fig. 1: “a minimum of 1000ms and in any case after manual encoding of the participant’s response by the experimenter.” Does this mean that the experimenter initiated each trial, e.g., by a button press? If yes, this should be stated in the manuscript explicitly.

Reply. After the participant’s vocal response, the experimenter encoded the response, and then there was an ITI of 1000ms before the beginning of the next trial. We have clarified this point in the procedure (lines 278-279).

Reviewer 2, point 11. 11. Through the text, the authors report SEM, even for descriptive statistics of accuracy (lines 289-291) or RT (Table 1). I am more familiar with reporting SDs in these contexts. Is there a particular reason for reporting SEM instead?

Reply. Considering that SEM refers to variability around a distribution of different means, with error rate (or accuracy) and reaction times we use SEM because the values we describe are the averaging of different means, each of them obtained by averaging many trials for each participant and condition. That said, as far as we know, both SEM and SD are commonly used.

Reviewer 2, point 12. 12. The authors introduce the term transport component, an average of grasping and pointing conditions (line 344). But tapping also includes hand movement to the object. It probably requires a better term or a justification for why tapping is not a part of the transport component.

Reply. In the tapping condition, differently from the pointing and the grasping ones, the participant’s forearm and hand laid on the table and gently moved up and down. Participants were encouraged to avoid huge movements. During tapping, the participant’s arm did not change (i.e. the participant did not move the arm), while the forearm could move just a little to accompany the hand. In this sense, the movement trajectory (up and down) was minimal as compared to point and grasp. Also, tapping was not directed toward a specific target position, differently from point and grasp. No object was presented during tapping. We have clarified this point in the manuscript, in line with the reviewer’s suggestion (lines 252-261).

Reviewer 2, point 13. 13. It remained unclear to me how the clusters (Fig. 3c and 3d) were defined. Was a cluster analysis applied, or was a particular numerical threshold introduced on one or both variables? Since some observations from different clusters overlap on each variable, I assume the former. If this is the case, the clustering method should be described in detail. These figures are pretty complicated and are not referred to in Discussion. Perhaps, they could even be omitted in the manuscript if they do not add substantial evidence in favor of any of the hypotheses.

Reply. No automatic clustering algorithms were applied. Fig.3c and 3d are scatterplots of the slopes of the distance effect for small digits at the individual level. They show the variability of the observed effects. Specifically, the x axis shows individual slopes for small digits in the grasping (Fig.3c) and pointing (Fig.3d) conditions, while the y axis shows the mean between the slopes in the other conditions, where the mean is computed at the individual level. Groups differing in colours are defined by comparing the x and y values at the individual level: a participant showing a larger distance effect (i.e., more negative slope) after grasping (Fig.3c) or after pointing (Fig.3d) as compared to the distance effect in the other conditions (mean slope) falls in one group, while a participant showing a smaller distance effect (i.e., less negative slope) after grasping (Fig.3c) or after pointing (Fig.3d) as compared to the distance effect in the other conditions (mean slope) falls in the other group. For example, if a participant has a slope = -25 after grasping and a mean slope = -20 in the other conditions, these values will be represented as a red dot in Fig3c. If a participant has a slope = -18 after grasping and a mean slope = -22 in the other conditions, these values will be represented as a grey cross in Fig3c. If a participant has a slope = -2 after pointing and a mean slope = -21 in the other conditions, these values will be represented as a blue triangle in Fig3d. If a participant has a slope = -10 after pointing and a mean slope = -9 in the other conditions, these values will be represented as a grey cross in Fig3d. Importantly, Fig.3c and 3d provide a graphical illustration of the prevalence of the effects as reported in the manuscript. We agree with the reviewer that the graph required clarification to be understood easily. However, we consider important to show the results at the individual level, therefore we opted for keeping these graphs and providing a clearer description. Also, we briefly discussed the variability in the observed effects in the discussion section (lines 525-527).

Reviewer 2, point 14. 14. The authors hypothesize that ordinality is acquired earlier than cardinality (lines 440-443). I would refer here to a model of early mathematical development by Krajewski, which supports this idea. See Fig. 1 at: Krajewski, K., & Schneider, W. (2009). Early development of quantity to number-word linkage as a precursor of mathematical school achievement and mathematical difficulties: Findings from a four-year longitudinal study. Learning and instruction, 19(6), 513-526.

Reply. We thank Reviewer 2 for this suggestion, which we have integrated in the manuscript.

Reviewer 2, point 15. 15. In Conclusion, the authors claim that the study supports bidirectionality of action-number links (line 476). However, bidirectionality implies that numbers also influence actions. The authors did not test this direction in their study. I would remove this term.

Reply. Done.

Reviewer 2, point 16. 16. I would recommend the authors summarize once again the main findings of the study in one or two sentences in Conclusion, right after the first sentence (line 477), and before discussing methodological advances and suggestions for future research.

Reply. Done.

Reviewer 2, point 17. 17. I personally found it confusing to see a citation at the end of the conclusion, especially the one that includes “see also”. It makes an impression of an incomplete text. I would omit the last citations entirely, or if it is crucial for the authors to keep them, rephrase the sentence so that the references appear at its beginning.

Reply. We have removed the last citation because it was not crucial.

Reviewer #3

Reviewer 3, point 1. The authors investigated whether repetitive hand action affects number processing. The originality of the study lies in the fact that they studied the interaction between action and numbers with functionally and temporally dissociated tasks. The methods are sound (I only have one concern reported below) and the manuscript is overall well written. I would however like the authors to precise their hypotheses and interpretations and centre the introduction on their research question and what they can tell about their results. More precisely, my concerns/questions are:

-I am not convinced that the first part of the introduction on finger use is relevant for the paper (certainly not in these proportions). I understand that the link between finger use and number processing is more eloquent than the link between action (grasping, pointing) and number processing. However, finger use and actions directed towards objects are fundamentally different and their link with number processing might be too. As said by the authors, the link between numbers and fingers might arise from finger counting habits, while the link between numbers and action might arise from the common need to process magnitude (object size, distance and number magnitude). An alternative might be that number is grounded in hand sensorimotor experience, but I am not convinced that the current results on action allow drawing conclusions on the link between hand representation and number in a more general way (which, by the way, the authors didn’t do). If, however, the authors wanted to go this way, I would advise them to make clearer hypotheses and discuss their results in this framework. Here, for instance, they make the hypothesis that finger tapping should have no effect on subsequent number processing, which seems to me in contradiction with number processing being grounded in hand sensorimotor experience. Otherwise, I would suggest focusing the introduction on the link between number and action to go straight to the point and not lose the reader along the way.

Reply. We thank the reviewer for this thoughtful comment. The aim of this study was indeed to investigate the link between number and action, based on the idea that number and action share common magnitude-related processes. At a more general level, this idea is usually considered in agreement with embodied and grounded cognition theories, suggesting that number – or, in general, cognition – is grounded into sensorimotor experience. However, in the introduction we wanted to provide the reader with a comprehensive overview of the literature indicating the existence of links between numerical processing and the use of the hands, either for counting, or for grasping. Also, both types of hand-number links (finger counting and grasping-number interactions) might be built upon sensorimotor neural networks contributing to both hand processing and number processing. This common neural substrate might favour the development of hand-based number-related cultural habits, which in turn consolidate and shape the link between hand actions and numerical processing. That said, we have followed the reviewer’s suggestion, and we have reduced the discussion on finger counting, while also clarifying the use of tapping as control condition (see also our response to Reviewer 2’s point 12).

Reviewer 3, point 2. Most effects of number processing on action (or conversely) are « lateralized »: small/large numbers induce leftwards/rightwards deviations in hand trajectory and increase/decrease hand opening (or conversely). In the present paper, the authors choose to test the effect of action on number processing by using centralised movement without taking into account grip opening. May the authors explain this choice? Could they also discuss how this could explain their results?

Reply. The use of non “lateralized” movements, together with the temporal dissociation between the motor and the numerical task, is one of the central aspects of our paradigm. The reviewer is right that we had not clearly explained this point in the manuscript. As we discuss in the introduction (lines 143-155), lateralized effects as shown by previous studies have been often interpreted as evidence of common processing between number and hand action. While agreeing in considering these studies as seminal in the context of studies on embodied numerical processing, we however argue that the use of “lateralized” (or “categorized”) movements does not permit to disentangle between effects arising from common cognitive processing and effects arising from strategies - or ad-hoc mental representations – adopted to facilitate the execution of the task (e.g., strategies creating associations between small and grip closing, and right and grip opening, or associations between precision grip and small number, and power grasp and large number). In other words, there might be the possibility that – at least part of – these effects are due to the use of specific lateralised paradigms; embodied cognition refers to these aspects as “situated”. On the contrary, the strength of our paradigm lies in the absence of any potential lateralised effect, and on the temporal dissociation between the motor and the number task. In this way, the observed results can be more likely attributed to shared cognitive mechanisms between number and hand action: the effects of motor adaptation are observed in the number task when the adapted cognitive mechanisms are also commonly involved in numerical processing. Considering the reviewer’s comment, we have clarified this point in the manuscript (pages 7-9).

Reviewer 3, point 3. From the papers cited by the authors (e.g., Badets et al., 2010) it is not clear why the authors hypothesized that grasp preshaping and reaching would affect number processing differently. Could they clarify which previous results could predict that grasping, but not reaching, could affect small numbers or number distance (or something like that)? Please also clarify how they could be different (without necessarily specifying the direction), but the expected effects remain a bit obscure to me.

Reply. Hypothesis 2 concerns the idea that numerical processing, hand reach and grasp might specifically share common cognitive mechanisms for locating items in space and for estimating magnitudes. While locating items is necessary to define reach trajectory, estimating magnitude is required to select the appropriate grip aperture during grasping. We make the hypothesis that these same mental computations can be at play also during numerical processing, acting to localise numbers and estimate numerical magnitudes in a mental number space (see also our reply to reviewer 2’s point 6). Following this reasoning, we predict that the numerical processing could be independently affected by arm transport and by grip pre-shaping. To test this prediction, we analysed both the distance and the magnitude effects, which are commonly considered evidence of semantic processing of numbers. More precisely, we wanted to test the hypothesis that adaptation to reaching (i.e., arm transport common to grasp and point actions) could bias the distance effect, because of a common mechanism of spatial localisation, while adaptation to grasp preshaping (specific of the grasp action) could bias the magnitude effect, because of a common mechanism for size estimation. However, we should acknowledge that these specific predictions are a consequence of our theoretical idea of shared common cognitive mechanisms for locating items in space and for estimating magnitudes, while previous studies were based on different rationales. In the current version of the manuscript, we have extensively clarified our hypothesis n.2, as well as its explorative nature.

Reviewer 3, point 5. Please note that Geers et al. (2021) also showed an effect of a functionally unrelated (but, concurrent) action on number comparison.

Reply. We thank the reviewer for this comment. We have reported the study by Geers et al. (2021) in the manuscript, which is indeed relevant in this context.

Reviewer 3, point 6. In the fine-grain analysis of the distance effect, the difference between grasping and the ensemble of the other conditions seems to be driven by the small distance effect in the pointing condition. Wouldn’t it be more relevant to compare grasping to each of the other conditions? It would guess that only the difference with pointing would be different, supporting a specific effect of pointing.

Reply. In the fine-grained analysis of the distance effect we compared grasping and pointing with the ensemble of the other conditions to be more consistent both with our predictions, and with the results of the preregistered ANOVA: specifically, the comparison between grasping vs. the ensemble of the other conditions allowed to test the effects of hand-preshaping, while the comparison of pointing vs. the ensemble of the other conditions allowed to test the specific effect of pointing as emerged from the ANOVA. That said, we have run the analyses suggested by the reviewer, and the results indicate that the size of the distance effect differed only between grasping and pointing (BF=14; p=.003). The other comparisons led to unclear results (BF<3 and >.3, therefore neither favouring the null hypothesis, nor the alternative hypothesis; all p-values >.01, i.e., not significant after applying Bonferroni correction). We have added these analyses in the manuscript and discussed possible explanations for these null findings. Specifically, while these results need to be replicated in future studies, future studies should also consider the use of different control conditions. In fact, both observation and tapping are not free from sensorimotor processing, with observation activating object affordances, and tapping being subserved by sensorimotor processing, albeit different from reach and grasp (as also pointed out by the reviewer in point 1). 

Reviewer 3, point 7. Might the authors develop a bit more there how their results support a specific link between grasp and cardinality and point and ordinality?

Reply. Also following the reviewer 2’s suggestion (points 6), we have further developed our interpretation of the findings in the discussion section and clarified why the results might support a specific link between grasp and cardinality and point and ordinality (lines 498-510).

Reviewer 3, point 8. Unless I missed it, the link to access the data on OSF has not been provided.

Reply. The reviewer is right. Our apologies for this error. We have now added the link to access the data (Data availability statement, link: https://osf.io/a8jzp/).

---

## [Decision Letter · Decision Letter 1]

24 May 2022

Influences of hand action on the processing of symbolic numbers: a special role of pointing?

PONE-D-21-36636R1

Dear Dr. Ranzini,

We’re pleased to inform you that your manuscript has been judged scientifically suitable for publication and will be formally accepted for publication once it meets all outstanding technical requirements.

Kind regards,

Jérôme Prado

Academic Editor

PLOS ONE

Additional Editor Comments (optional):

Reviewers' comments:

Reviewer's Responses to Questions

**Comments to the Author**

1. If the authors have adequately addressed your comments raised in a previous round of review and you feel that this manuscript is now acceptable for publication, you may indicate that here to bypass the “Comments to the Author” section, enter your conflict of interest statement in the “Confidential to Editor” section, and submit your "Accept" recommendation.

Reviewer #3: All comments have been addressed

2. Is the manuscript technically sound, and do the data support the conclusions?

Reviewer #3: Yes

3. Has the statistical analysis been performed appropriately and rigorously? 

Reviewer #3: Yes

4. Have the authors made all data underlying the findings in their manuscript fully available?

Reviewer #3: Yes

5. Is the manuscript presented in an intelligible fashion and written in standard English?

Reviewer #3: Yes

6. Review Comments to the Author

Reviewer #3: The authors have done a remarkable job answering my comments. The changes have really improved the manuscript. I have no further comments.

7. PLOS authors have the option to publish the peer review history of their article (what does this mean?). If published, this will include your full peer review and any attached files.

Reviewer #3: **Yes: **Laurie Geers

---

## [Editor Report · Acceptance letter]

2 Jun 2022

PONE-D-21-36636R1 

Influences of hand action on the processing of symbolic numbers: a special role of pointing? 

Dear Dr. Ranzini:

I'm pleased to inform you that your manuscript has been deemed suitable for publication in PLOS ONE. Congratulations! Your manuscript is now with our production department. 

Kind regards, 

on behalf of

Dr. Jérôme Prado 

Academic Editor

PLOS ONE